# The Effects of Co on the Microstructure and Mechanical Properties of Ni-Based Superalloys Prepared via Selective Laser Melting

**DOI:** 10.3390/ma16072926

**Published:** 2023-04-06

**Authors:** Xiaoqiong Ouyang, Feng Liu, Lan Huang, Lin Ye, Heng Dong, Liming Tan, Li Wang, Xiaochao Jin, Yong Liu

**Affiliations:** 1State Key Laboratory of Powder Metallurgy, Central South University, Changsha 410083, China; 2Powder Metallurgy Research Institute, Central South University, Changsha 410083, China; 3Foshan (Southern China) Institute for New Materials, Foshan 528200, China; 4State Key Laboratory for Strength and Vibration of Mechanical Structures, Xi’an Jiaotong University, Xi’an 710049, China

**Keywords:** Ni-base superalloy, selective laser melting, cobalt, strength, ductility

## Abstract

In this work, two Ni-based superalloys with 13 wt.% and 35 wt.% Co were prepared via selective laser melting (SLM), and the effects of Co on the microstructure and mechanical properties of the additively manufactured superalloys were investigated. As the Co fraction increased from 13 wt.% to 35 wt.%, the average grain size decreased from 25.69 μm to 17.57 μm, and the size of the nano-phases significantly increased from 80.54 nm to 230 nm. Moreover, the morphology of the γ′ phase changed from that of a cuboid to a sphere, since Co decreased the γ/γ′ lattice mismatch from 0.64% to 0.19%. At room temperature, the yield strength and ultimate tensile strength of the 13Co alloy reached 1379 MPa and 1487.34 MPa, and those of the 35Co alloy were reduced to 1231 MPa and 1350 MPa, while the elongation increased by 52%. The theoretical calculation indicated that the precipitation strengthening derived from the γ′ precipitates made the greatest contribution to the strength.

## 1. Introduction

Ni-based superalloys have been broadly applied in aerospace industries due to their excellent high-temperature properties, such as their high strength and oxidation resistance [1,2]. In general, they are often used for hot-section components of engines, such as burners, turbine discs, turbine blades, etc. [3]. Significantly, the quality of a turbine engine is closely related to improvements in Ni-based superalloy performance. Traditionally, superalloy components were prepared by using casting, forging, and hot extrusion [4]. In contrast, these methods have disadvantages, such as complex preparation processes, low efficiency, and high costs [5], making it difficult for them to meet the requirements for forming integrated components with complex structures. The selective laser melting (SLM) technology can be used to directly shape powder particles, and the laser can be changed according to the geometry of the products. Hence, the preparation process for SLM is relatively simple, and it can be suitable for the preparation of complex structural parts. At the same time, it can ensure the dimensional accuracy of a product [6] and a good surface roughness [7].

With the application of SLM in Ni-based superalloys, researchers have tried to change the internal microstructures of the alloys by adjusting the elemental content to enhance the mechanical properties. Griffiths et al. [8] found that removing the Hf element could reduce the number of cracks and improve the yield strength of low-carbon CM247LC superalloys. In low-carbon IN738LC, the ultimate tensile strength was increased from 610 to 1113 MPa when the Zr content was reduced from 0.12 wt.% to 0.024 wt.% [9]. In addition, by increasing the Co fraction from 5 wt.% to 23 wt.%, Ni-Co-based superalloys exhibited a higher yield strength and ductility at 750 °C and 800 °C [10]. Murray et al. [11] obtained a crack-free Co-Ni-based superalloy with a high ultimate tensile strength of 1.1 GPa and elongation of over 13% at room temperature by increasing the Co content to 39 wt.%. Based on these studies, it is expected that the mechanical properties of additively manufactured Ni-based superalloys could be enhanced by adjusting the Co content.

Co can substitute Ni in the crystal lattice, and it is broadly deemed one of the main elements in γ′-strengthened Ni-based superalloys. Based on previous works, it was found that Co can cause γ′-Ni_3_(Al,Ti) to transform into γ′-(Ni,Co)_3_(Al,Ti) [12,13,14], reduce the stacking fault energy, and modify the lattice mismatch of γ/γ′ to influence the morphology of γ′ precipitates [15,16]. However, the effects of Co on other precipitates, such as carbides and the σ phase, have not yet been studied. Actually, except for γ′ precipitates, factors originating from other precipitates, i.e., their morphology, size, distribution, and stability in superalloys, also have an important influence on the mechanical properties [17,18,19].

In this research, two superalloys with different Co contents were prepared via selective laser melting, and the γ′ precipitates, carbides, and σ phase formed in the alloys were characterized. The effects of Co on the morphology and size of the precipitates were systematically analyzed, and the contributions to the strength from different strengthening mechanisms are discussed in detail.

## 2. Materials and Methods

### 2.1. Materials

γ′-strengthened Ni-based superalloys with different Co contents were the research object of the experiments. Table 1 shows the nominal chemical compositions of two pre-alloyed powders with Co contents of 13 wt.% and 35 wt.%. Both powders were produced by using gas atomization, and the distribution range of two particles’ sizes was 10–63 μm, with D_50_ being around 34.3 μm, where D_50_ means that nearly half of the powder particles exceeded the particle size value that it represents. Figure 1a,b show the morphology of the powders. There were some satellite powders for 13Co, and the powder distribution was more uniform than that of 35Co.

### 2.2. SLM Process

The Farsoon 271M SLM device was used to prepare rectangular samples with dimensions of 28 × 12 × 14 mm^3^. Argon was used as a protective atmosphere during the printing process. The following process parameters were used for preparation: a laser power of 200 W, a scanning speed of 1200 mm/s, a hatch space of 80 μm, and a layer thickness of 40 μm. Stainless steel material was used as the substrate and was preheated to 100 °C before printing. The strategy of bidirectional scanning with an interlayer rotation of 67°, as illustrated in Figure 1c, was chosen to reduce the residual stress between the layers [20].

### 2.3. Heat Treatment Process

The as-printed samples were heated at 1180 °C for super-solution treatment for 1 h and then set in the air to cool to room temperature. After that, aging treatments were conducted in two stages: first, the samples were heated at 650 °C for 24 h, followed by allowing them cool down in the air; second, the samples were deposited at 760 °C for 16 h, and then allowed to cool down in the air.

### 2.4. Microstructural Characterization

Archimedes’ principle was used to measure the relative density of the as-printed samples. Before microstructural characterization, SiC papers were used to sand the surfaces of the samples. Subsequently, all samples were polished to obtain mirror-like surfaces. All samples were etched with a Kalling reagent consisting of 50 g of CuCl_2_, 100 mL of HCl, and 100 mL of methanol for observation with scanning electron microscopy (SEM, FEI Quanta 650 FEG, Tescan, Brno, Czech Republic). An electron backscattered diffraction (EBSD) investigation was undertaken at a voltage of 20 kV, a magnification of 500, and a step size of 2 μm. The data were processed by using the Channel 5 software. An X-ray diffraction (XRD) investigation was conducted on a Bruker D8 Advanced X-ray diffractometer(Bruker, Saarbrücken, Germany) with a scanning speed of 5°/min to measure the phase constitution and lattice parameters of the samples after heat treatment. The heat-treated samples were ground to a thickness of 50 μm and then subjected to ion thinning for transmission electron microscopy (TEM) observations. The TEM observations were conducted by using a Thermo Scientific™ Titan Spectra 300(Thermofisher, New York, NY, USA) equipped with a quadrant Super-X detector. The high-resolution scanning TEM (HRSTEM) and EDS results were collected and analyzed by using the Thermo Scientific™ Velox software.

### 2.5. Mechanical Properties

Tensile tests were conducted on the UTM5105 electronic universal testing machine. Tensile specimens were cut from rectangular samples by using the wire-cutting technique, and the profile of the tensile model is illustrated in Figure 1d. All tests were executed according to the ASTM E8 and E21 standards. The size of all tensile specimens in this work was 26 × 10 × 2 mm^3^. The tensile tests were conducted at 25 °C and 850 °C with a strain rate of 10^−3^ s^−1^. All tensile tests were repeated three times.

## 3. Results and Discussion

### 3.1. As-Printed Microstructure

The relative densities of the as-printed13Co and 35Co samples as measured with Archimedes’ drainage method were both 99.6%, which indicated that the degrees of internal defects in 13Co and 35Co were the same. The backscattered electron (BSE) images of the XZ planes in the as-printed 13Co and 35Co alloys are illustrated in Figure 2a,b, respectively. Although the laser power, scanning speed, and hatch space were optimized to some extent, some cracks still appeared along the melt pools, and a few holes existed. Melt pools appeared in the XZ plane in the form of a “V” shape, and Figure 2c,d show that the melt pools in the XY plane were elliptical. The morphology of the melt pools was mainly related to the interaction between the laser and the powder [21] during the preparation of SLM. The absorption rate of the powder with the laser determined the degree of the laser’s penetration of the powder and the depth of the melt pools, thereby affecting the morphology of the melt pools [22].

### 3.2. Grains

As shown in the inverse pole figures (IPFs) shown in Figure 3a,b, the grain size in the XY plane of the 13Co alloy ranged from 3 to 60 μm, and the average grain size was about 8 μm. The situation was similar in the 35Co alloy, where the range of the grain size in the XY plane was 2–63 μm, and the average grain size was about 8.96 μm. Thus, the difference in grain size between the two as-printed alloys was insignificant. Additionally, this displayed that the XY planes of both alloys presented the phenomenon of “small grains surrounding a large grain”, which was mainly caused by the laser-scanning strategy with an interlayer rotation of 67°. It is worth mentioning that there were significantly more small grains around the large grains in the 35Co alloy compared to the 13Co alloy.

After the super-solution treatment and aging treatments in two stages, the grains of the 13Co and 35Co alloys became coarse. However, the grains of the 35Co alloy were finer than those of the 13Co alloy, and the phenomenon of “small grains surrounding large grains” disappeared in the XY plane for both alloys, which can be observed in Figure 3c,d. Twins could be discerned, as indicated by the white circles in the images, and more twins were observed in 35Co than in 13Co, since the stacking fault energy of the γ matrix decreased with the increase in Co content [23]. After heat treatment, recrystallization occurred, and the fractions of the recrystallized grains for the 13Co and 35Co alloys were obtained by using the Recrystallized Fraction Component Function in the Channel 5 software, as displayed in Figure 4. It can be seen in Figure 4a,c that the deformation degree of 13Co before heat treatment was larger than that of 35Co. After heat treatment, in the 13Co alloy, the fraction of the recrystallized grains accounted for above 90%, as shown in Figure 4b, while it was only close to 50% in the 35Co alloy, as shown in Figure 4d. This demonstrated that the recrystallization of the 13Co alloy might have been close to completion, while the recrystallization process still took place in the 35Co alloy.

According to previous studies [24,25], the greater the degree of deformation, the finer the grains will be after heat treatment. However, the grains of the 35Co alloy after heat treatment were finer than those of the 13Co alloy. Comparing Figure 3a,b, it can be seen that there were many small grains in the as-printed 35Co alloy, and more significant grain boundaries can provide more positions for nucleation. However, the internal deformation of 35Co was less than that of the as-printed 13Co alloy. Hence, the deformation energy could not provide the required driving force, resulting in finer grains after heat treatment.

### 3.3. Precipitate Distribution

#### 3.3.1. XRD Analysis

The XRD spectra of the 13Co and 35Co alloys after heat treatment are shown in Figure 5, illustrating that the 13Co and 35Co alloys were substantially the same in terms of their phase configuration. The diffraction peaks in 35Co were all slightly shifted to the left, which was mainly caused by the increase in the Co content, since it increased the lattice parameters of γ and γʹ. It should be noted that no reflections from other phases could be revealed in the XRD analysis, which was probably due to their small fractions. Thus, further electron microscopy investigations were conducted for the investigated alloys.

#### 3.3.2. Formation Mechanisms and the Difference in the Sizes of MC Carbide and the σ Phase

Figure 6a–d clarify that SEM was used to further examine the microstructures of the 13Co and 35Co alloys after heat treatment in the backscattered electron mode; precipitates formed in both alloys, as shown with the white contrast. The white precipitates randomly existed in the samples. In addition, they were more prominent in the 35Co alloy than in the 13Co alloy. SEM images recorded in five randomly selected regions were used to evaluate the average size and volume fraction of these white precipitates, which were determined to be 81 nm and 1.3% in 13Co and 230 nm and 2.1% in 35Co, respectively.

The white precipitates in the two alloys were further characterized by using TEM, as shown in Figure 7. From the high-angle annular dark field-scanning transmission electron microscope (HAADF-STEM) micrograph and EDS maps of the 13Co alloy (Figure 7a), it can be inferred that the white precipitates were composed of two different phases, which was mainly because the elements Ta, Ti, and C were concentrated on one white precipitate and Mo and W were concentrated on the other white precipitate. A similar phenomenon appears in Figure 7b, so it can be tentatively judged that the types of the white precipitates were the same in 13Co and 35Co alloys.

Based on the EDS mapping, the white precipitates in 13Co and 35Co were probably the same. Figure 8 shows a further investigation of the two white precipitates in the 13Co alloy with high-resolution TEM to clarify their structures. The existence of the standard FCC-structured MC-phase carbide shown in Figure 8a was confirmed by the high-resolution image in Figure 8c and the associated fast Fourier transform (FFT) pattern along the axis of the [0,$\overset{-}{1}$,1] zone. Figure 8b exhibits the formation of the tetragonally structured σ phase, which was confirmed by the high-resolution image and the associated FFT pattern along the axis of the [$\overset{-}{1}$,0,0] zone shown in Figure 8d.

Through comprehensive investigations with SEM and TEM, the white precipitates appearing in the two alloys were identified as MC carbide and the σ phase. The formation of MC carbide could be attributed to the segregation of Ta, Ti, and C during the final solidification stage [26]. In Ni-based superalloys, Ta and Ti have stronger affinity with C than other elements do [27,28], which can prevent the combination of other elements with C, which would result in the formation of different kinds of carbides and the inhibition of the decomposition of MC carbides. It may be that the consumption of γ′ precipitates relative to the Al and Ti elements was less significant, thereby promoting the formation and growth of MC carbides.

Mo and W are the main elements that formed the σ phase, as this is one of the TCP phases [29,30]. Sites with defects tend to favor the nucleation position of the σ phase, which is related to the low activation energy around them [31]. The ranking of defects in order of increasing activation energy is as follows: grain boundaries < twin boundaries < dislocations [31,32]. Therefore, the σ phase is difficult to nucleate and grow in recrystallized grains. Due to the mass of dislocations inside the non-recrystallized grains, many nucleation checkpoints can be provided for the σ phase [33]. The growth of the σ phase is mainly affected by atomic diffusion, which often needs grain boundaries and twin boundaries to provide paths. By comparing the recrystallization profiles of the 13Co and 35Co alloys, it could be seen that the number of non-recrystallized grains in 35Co after heat treatment was relatively high, so there were many nucleation checkpoints for the σ phase, and the number of grain boundaries and twin boundaries in 35Co was more significant, so it could provide favorable diffusion paths for the growth of the σ phase.

### 3.4. Transformation Mechanism of the γ′ Morphology

Figure 9 shows HAADF-STEM images of the γ′ precipitates and corresponding EDS maps in the 13Co and 35Co alloys. The γ′ precipitates could not be quickly revealed in the HAADF-STEM images, but could be confirmed by the element maps. It could be seen that Ni, Al, Ti, and Ta were enriched in the γ′ precipitates, and Co, Cr, and Mo were enriched in the γ matrix. By analyzing five SEM images that were recorded in randomly selected regions of both alloys by using Image J, the volume fractions of the γ′ precipitates were determined to be about 48% and 38%. Figure 9a reveals that the γ′ morphology was cubic in the 13Co alloy, and the average size was about 239 nm. However, the γ′ morphology in the 35Co alloy was spherical, as shown in Figure 9b, and the average size was about 187 nm. It can be concluded that there were obvious differences in the morphologies of the γ′ precipitates.

Some studies have shown that the morphology of γ′ precipitates is mainly related to the γ/γ′ lattice mismatch [34,35,36]. Due to the difference in the atomic radii of Co and Ni, the addition of Co could affect the lattice parameters of γ and γ′, thereby affecting the lattice mismatch of γ/γ′. To obtain the influence on the lattice parameters of γ,γ′ and the lattice mismatch of γ/γ′, as shown in Figure 5, a Gaussian function was used to fit the (200)_γ/γ′_ peaks in the XRD patterns, and Figure 10 exhibits the fitting results. The lattice mismatch of γ/γ′ was then calculated with the following equations:
(1)$$a=\frac{\lambda }{2\mathrm{Sin}\theta }\sqrt{h^{2}+k^{2}+l^{2}}$$
(2)$$\delta =\frac{2\left(a_{\gamma \prime }-a_{\gamma }\right)}{a_{\gamma \prime }+a_{\gamma }}$$
where *a*_γ_ and *a*_γ′_ are the lattice parameters of the γ and γ′ phases, respectively. The lattice parameters of γ/γ′ in 13Co and 35Co were calculated to be 0.3596/0.3619 nm and 0.3597/0.3604 nm, respectively, and the lattice mismatches of γ/γ′ were 0.64% and 0.19%, respectively. Although the lattice mismatch of γ/γ′ in 13Co exceeded 0.5%, it still remained cubic, which is considered normal [37]. Therefore, the morphology of the shift of the γ′ phase from cubic to spherical resulted from the increase in Co content.

### 3.5. Mechanical Properties

#### 3.5.1. Fractured Morphology

The tensile properties of the heat-treated samples were tested at room temperature and 850 °C. The tensile results of all specimens are shown in Figure 11. At room temperature, the yield strength, ultimate tensile strength, and elongation were 1379 MPa, 1487 MPa, and 5.75% for the 13Co alloy and 1231 MPa, 1350 MPa, and 8.75% for the 35Co alloy, respectively. At 850 °C, the ultimate tensile strengths of the 13Co and 35Co alloys were around 329 MPa and 412 MPa, and their elongations were 2% and 2.25%, respectively. Compared to the 13Co alloy, the strength was increased by 25.5% and the ductility was slightly increased for the 35Co alloy. It is worth noting that the SLM 13Co and 35Co alloys after heat treatment showed better mechanical properties than those reported for Ni-based superalloys. Figure 12 shows the relationship between the yield strength and (Al + Ti) content. Based on the simple principle that alloys with (Al + Ti) content exceeding 6% are nonweldable [38] and a yield strength over 1200 MPa is deemed high, Figure 12 is divided into four regions.

The fractured characteristics of the tensile specimens of 13Co and 35Co at room temperature are exhibited in Figure 13. Figure 13a,b demonstrate that there were some microcracks and pores on the fracture surface. Microcrack tips often acted as stress concentration points, causing accelerated crack propagation with extremely adverse effects on the tensile properties. In addition, a large number of dimples and intragranular tears were detected, as shown in Figure 13b,d. Judging from Figure 13c,f, the dimples of 35Co are smaller and deeper, indicating its better ductility than that of 13Co.

#### 3.5.2. Deformed Microstructures

In order to further understand the deformation of the structures during stretching, the side parts of the fractures were taken and characterized by using TEM. Figure 14a shows the stacking fault shearing of the γ′ phase in 13Co; a large amount of dislocation was plugged at the grain boundary. Generally, when the dislocations encountered the coarse γ′ particles, the shearing and dissociation of the dislocations usually caused stacking faults [10]. On the other hand, under stress, at the γ/γ′ interface, the ½[110] dislocation dissociated into two Shockley dislocations, where 1/6[112] Shockley partials would shear the γ′ precipitates, leading to the formation of stacking faults [49]. When a partial dislocation encountered a stacking fault, it could be hindered, resulting in T-shaped stacking faults. As shown in Figure 14b, there were full ½[110] dislocations in the matrix, and they often occurred as dislocation pairs at room temperature. Therefore, the ordered γ′ phase was sheared by weakly coupled dislocation pairs or strongly coupled dislocation pairs. There was a large number of dislocation cells, as shown in Figure 14c, and the formation of dislocation cells was mainly related to the large dislocation density and the accumulation of dislocations. In addition, the phenomenon of nanoparticle-pinning dislocations was also found, as shown in Figure 14d. Fine and dispersed nanoparticles were strong barriers for dislocation glide [18]. Since the nanoparticles could hinder the movement of dislocations, dislocation entanglement occurred at the interface of incoherent nanoparticles and the γ matrix, resulting in local stress concentration and plasticity reduction.

More stacking faults and T-shaped configurations can be observed in Figure 15a,c. A large number of dislocation pairs that sheared γ′ particles can be observed in Figure 15b; an APB-coupled dislocation pair that shears the γ′ phase is a common deformation mechanism in Ni-based superalloys [50,51]. Deformed microtwins can also be observed in Figure 15d. Microtwins and stacking faults caused by dislocation dissociation play a key role in the strengthening process of alloys. The contributions of different strengthening mechanisms to the strength of the two alloys will be shown in the next section.

#### 3.5.3. Effect of Microstructure on Mechanical Properties

Based on the previous microstructural analysis of 13Co and 35Co, the differences were mainly reflected in the γ′ phase, MC carbide, and σ phase. To explain the difference in mechanical properties between the two superalloys, a physical model was established to quantify the various effects on the yield strength of the alloys at room temperature. The yield strength *σ*_0.2_ is the sum of the strength contributions of each mechanism, mainly including solid solution strengthening *σ*_ss_, grain boundary strengthening *σ*_Gb_, γ′ phase strengthening *σ*_p_, and nano-phase strengthening *σ*_p-cut_.
(3)$$\sigma_{0.2}=\sigma_{Ss}+\sigma_{Gb}+\sigma_{P}+\sigma_{p-cut}$$

##### Solid Solution Strengthening

For γ′-precipitation-strengthened Ni-based superalloys, L. Gypen [52] et al. proposed an expression for calculating *σ*_ss_:(4)$$\sigma_{Ss}=\left(1-f_{\gamma \prime }\right){\left[\sum {\left(\beta_{i}x_{i}^{1/2}\right)}^{2}\right]}^{1/2}$$
where *f_γ′_* represents the volume fraction of the *γ′* phase. *x_i_* is the atomic percentage of the *i*-th element represented in the γ matrix, and *β_i_* represents the solid solution strengthening coefficient of the *i*-th element, which is related to the elemental atomic radius and modulus. The *β*_i_ coefficient was determined with reference to [53].

##### Grain Boundary Strengthening

The grain boundary strengthening of Ni-based superalloys follows the Hall–Petch relationship and can be expressed as:(5)$$\sigma_{Gb}=K_{Hp}d^{-1/2}$$
where *K_HP_* is an experimental constant related to the material properties and *d* is the average grain size. For superalloys, the value of *K_HP_* ranges from 710 to 750 MPa·μm^−1/2^ [50], and in this study, the *K*_HP_ value of 710 was used [54].

#### 3.5.4. Precipitation Strengthening

Precipitation strengthening is the most essential strengthening mechanism in Ni-based superalloys. The critical resolved shear stress (CRSS), which is defined as a driving force that can make dislocation pairs pass through the γ′ phase, is proportional to the precipitation intensity [54]. The precipitation strengthening can be divided into weak pair coupling, strong pair coupling, and the Orowan ring, which is based on the relationship between the size of the precipitates and the distances of the pairs of dislocations. According to Reed’s theory [55], the shear stress of weak pair coupling, strong pair coupling, and pair coupling can be described as
(6)$$\tau_{weak}=\frac{\gamma_{APB}}{2b}\left[{\left(\frac{6\gamma_{APB}fr}{2\pi T}\right)}^{1/2}-f\right]$$
(7)$$\tau_{strong}=\sqrt{\frac{3}{2}}\left(\frac{Gb}{r}\right)\frac{f^{1/2}}{\pi^{2/3}}{\left(\frac{2\pi \gamma_{APB}}{Gb^{2}}-1\right)}^{1/2}$$
where *b* and *G* represent the Burgers vector and shear modulus, respectively; this study used 0.254 nm [56] and 80 GPa [54,57,58]. *T* is the line tension of dislocation and is numerically equal to (*Gb*^2^)/2. *R* and *f* represent the average size and volume fraction of the γ′ phase. ***γ***_APB_ is the APB energy, which represents the threshold required for dislocations to shear through the precipitate. Furthermore, *γ*_APB_ can be expressed as follows [59]:(8)$$\gamma_{APB}=\gamma_{APB}^{0}+\sum_{i=1}^{n}k_{i}c_{i}$$
where *c_i_* is the atomic fraction of solute *i* in the γ′ phase, *n* is the number of solute elements in the γ′ phase, and $\gamma_{APB}^{0}$ = 195 mJ/m^2^ [60]. Here, the coefficient of APB energy change, *k*_i_, _is_ determined by density functional theory (DFT) with reference to [59].

When the size of the precipitates exceeds the critical radius of γ′, the dislocation can bypass the γ′ phase to form an Orowan ring, and the CRSS *τ_orowan_* of the Orowan ring is described as follows [61]:(9)$$\tau_{orowan}=\frac{3Gb}{2L}$$
(10)$$L={\left(\frac{2\pi }{3f}\right)}^{1/2}r$$
(11)$$\sigma_{P}=M\left(\tau_{weak}^{n}+\tau_{strong}^{n}+\tau_{orowan}^{n}\right)$$
where *M* is the Taylor factor, which was taken as 3.06 [57], and *n* is a fitting coefficient, which was taken as 5/6 in this work.

##### Nanoparticle Strengthening

According to the R-B model [62], precipitates will interact with a dislocation, thus hindering the movement of the dislocation and resulting in an increase in intensity [63]. Assuming that dislocations can be accordingly strengthened through weak particles, the expression is as follows [62]:(12)$$\sigma_{p-cut}=M\frac{Gb}{L}{\left[1-{\left(\frac{E_{p}}{E_{m}}\right)}^{2}\right]}^{3/4}$$
where *E_p_* represents the dislocation line energies in the nanoparticles and *E_m_* represents the dislocation line energies in the γ matrix. *L* is the average distance between nanoparticles in the glide plane. The *E_p_*/*E_m_* ratio is 0.62 [64].

In the 13Co alloy, the strength contribution of each part was $\sigma_{ss}$ = 220.85 MPa, $\sigma_{Gb}$ = 140.08 MPa, $\sigma_{p}$ = 916.69 MPa, and $\sigma_{p-cut}$ = 41.64 MPa; in the 35Co alloy, the strength contribution of each part was established to be $\sigma_{Ss}$ = 259.88 MPa, $\sigma_{Gb}$ = 169.38 MPa, $\sigma_{p}$ = 954.61 MPa, and $\sigma_{p-cut}$ = 18.65 MPa. To sum up, the results of the calculated yield strengths of the 13Co and 35Co alloys were in agreement with the experimental results. In the 13Co and 35Co alloys, precipitation strengthening contributed the most, followed by solid solution strengthening. This indicated that the size and volume fraction of γ′ precipitates had the greatest influence on the yield strength of the 13Co and 35Co alloys.

## 4. Conclusions

In this study, 13Co and 35Co alloys were successfully fabricated via SLM. The effects of Co on the microstructure and properties of the additively manufactured superalloys were systematically investigated, and the main conclusions can be summarized as follows.

(1)After heat treatment, the grain size of the 13Co alloy was coarser than that of the 35Co alloy, since the larger strain formed in the as-printed 13Co alloy could provide a driving force for grain growth.(2)MC carbides and the σ phase were observed in both 13Co and 35Co, and their size increased from 80.54 nm to 230 nm with the increase in Co content. The increase in Co reduced the consumption of γ′ relative to the Al and Ti content, thereby promoting the formation and growth of MC carbides. The size difference of the σ phase was mainly related to the large number of dislocations in the non-recrystallized grains after heat treatment.(3)As the Co content increased from 13 wt.% to 35 wt.%, the lattice mismatch of γ/γ′ decreased from 0.64% to 0.19%, and the morphology of the γ′ precipitates shifted from cubic to spherical.(4)At room temperature, the yield strength of 13Co was 10.7% higher than that of 35Co, but the ductility is relatively lower, since the 13Co alloy had more internal cracks and the dislocation entanglement and local stress at the interface of incoherent nano-particles and the γ matrix led to a decrease in the elongation.(5)The strengthening mechanism of the 13Co and 35Co alloys was dominated by precipitation strengthening, which contributed about 70% to $\sigma_{0.2}$, followed by solid solution strengthening.

## Figures and Tables

**Figure 1 materials-16-02926-f001:**
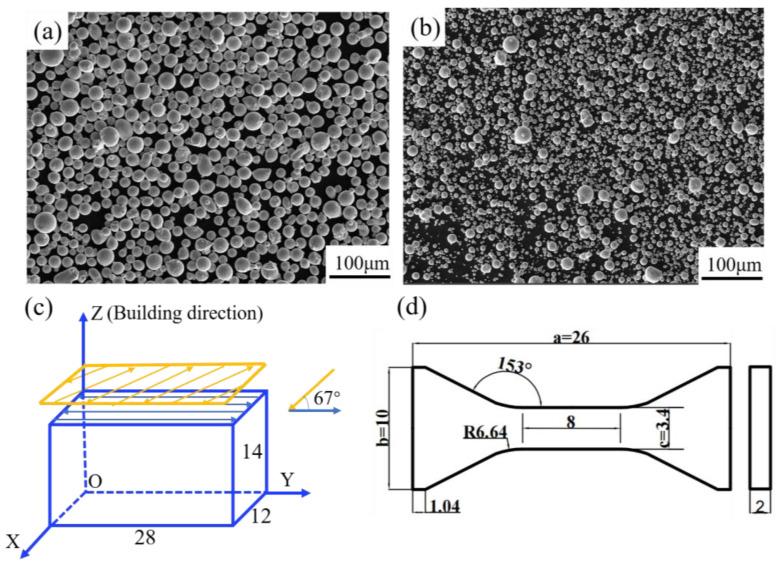
Secondary electron (SE) images of the powder morphology for (**a**) 13Co and (**b**) 35Co; the schematic images illustrate (**c**) the laser-scanning strategy and the size of a stretched specimen; (**d**) the geometry of the tensile specimen with a thickness of 2 mm.

**Figure 2 materials-16-02926-f002:**
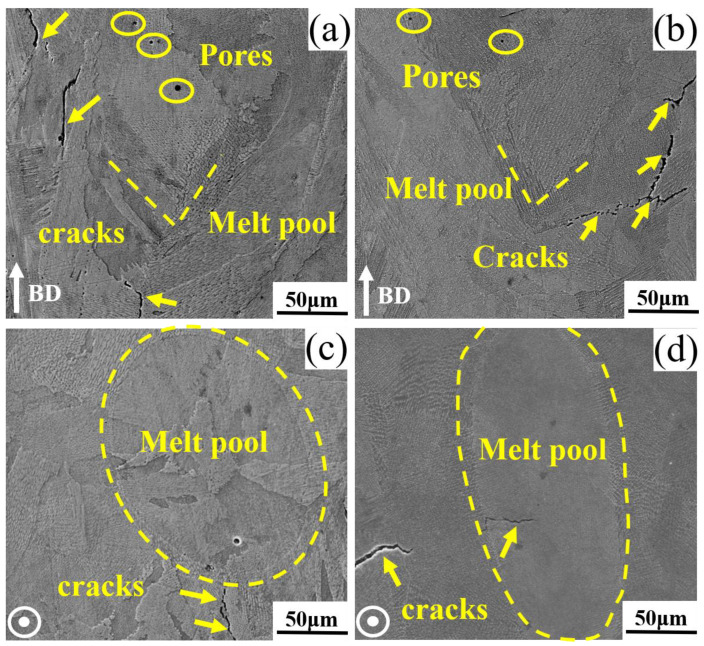
BSE images of as-printed microstructures in (**a**) the XZ plane in 13Co, (**b**) the XZ plane in 35Co, (**c**) the XY plane in 13Co, and (**d**) the XY plane in 35Co. The XZ plane was parallel to the building direction, and the XY plane was perpendicular to the building direction.

**Figure 3 materials-16-02926-f003:**
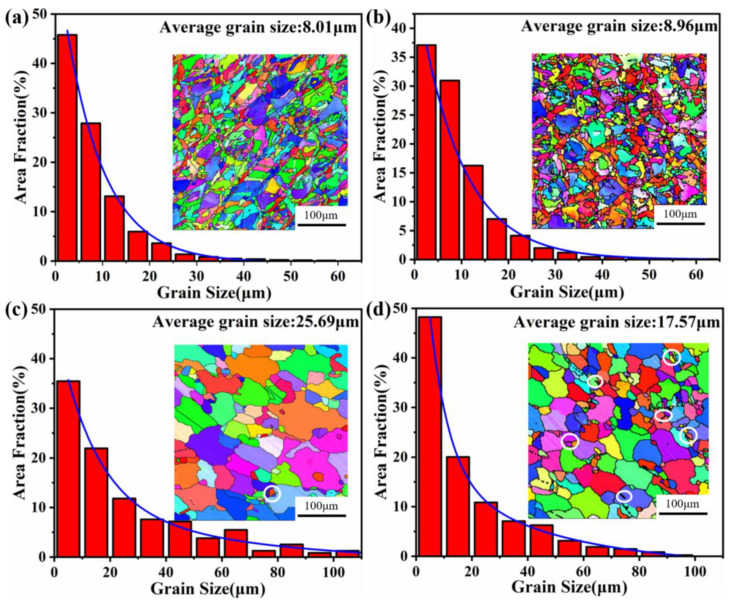
IPF maps obtained from the EBSD analysis in the XZ plane of the alloys and the corresponding grain size distribution diagram: (**a**) as-printed 13Co alloy; (**b**) as-printed 35Co alloy; (**c**) heat-treated 13Co alloy; (**d**) heat-treated 35Co alloy. The white circles represent twins.

**Figure 4 materials-16-02926-f004:**
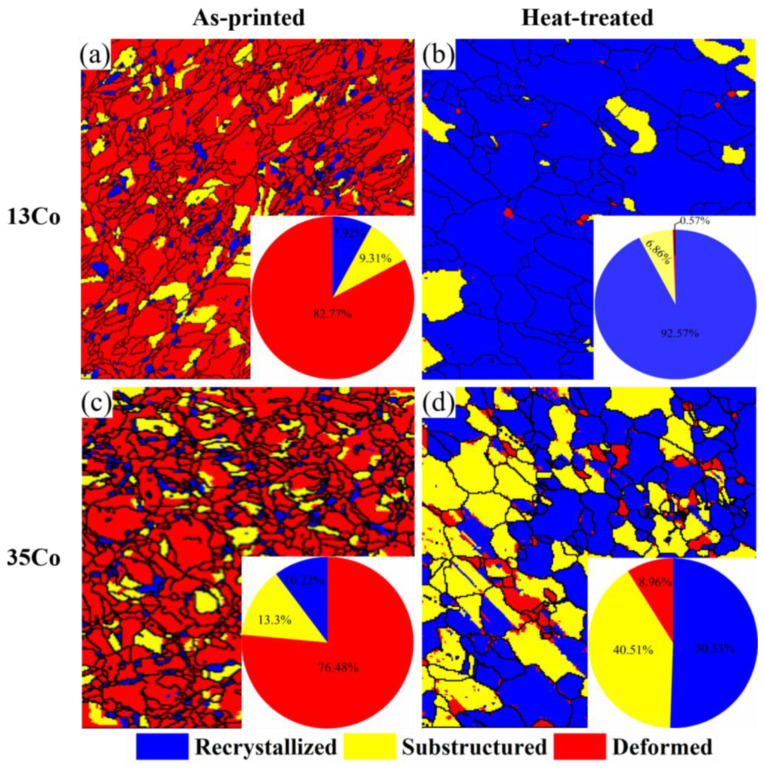
Recrystallization diagram of the XZ planes in (**a**) the as-printed 13Co, (**b**) heat-treated 13Co, (**c**) as-printed 35Co, and (**d**) heat-treated 35Co with recrystallized, substructured, and deformed grains.

**Figure 5 materials-16-02926-f005:**
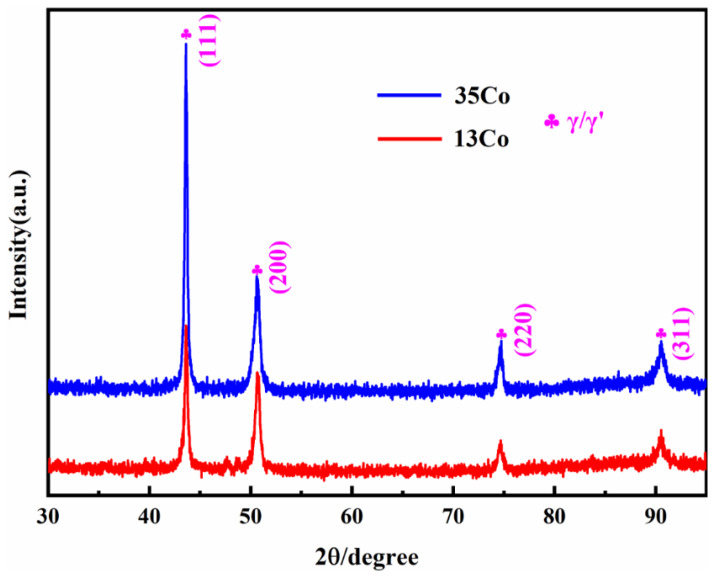
XRD spectra of the 13Co and 35Co alloys after heat treatment.

**Figure 6 materials-16-02926-f006:**
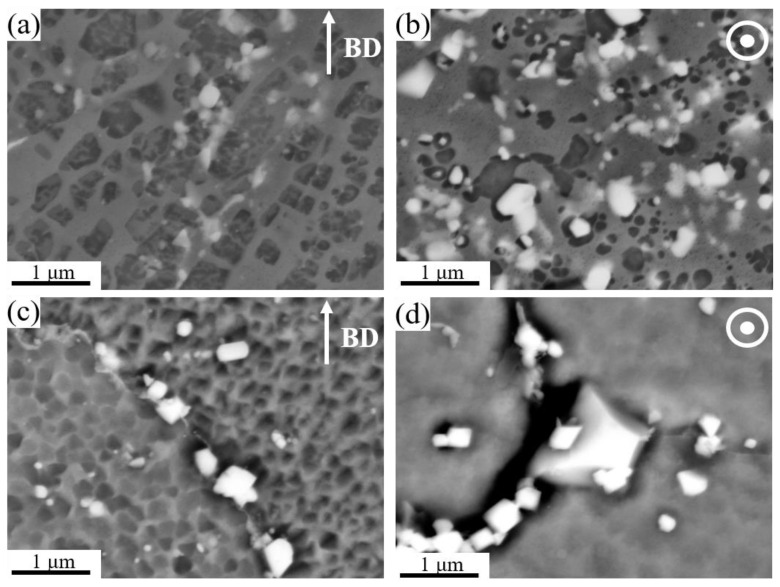
BSE images showing the microstructure after heat treatment of (**a**) the XZ plane of 13Co, (**b**) the XY plane of 13Co, (**c**) the XZ plane of 35Co, and (**d**) the XY plane of 35Co.

**Figure 7 materials-16-02926-f007:**
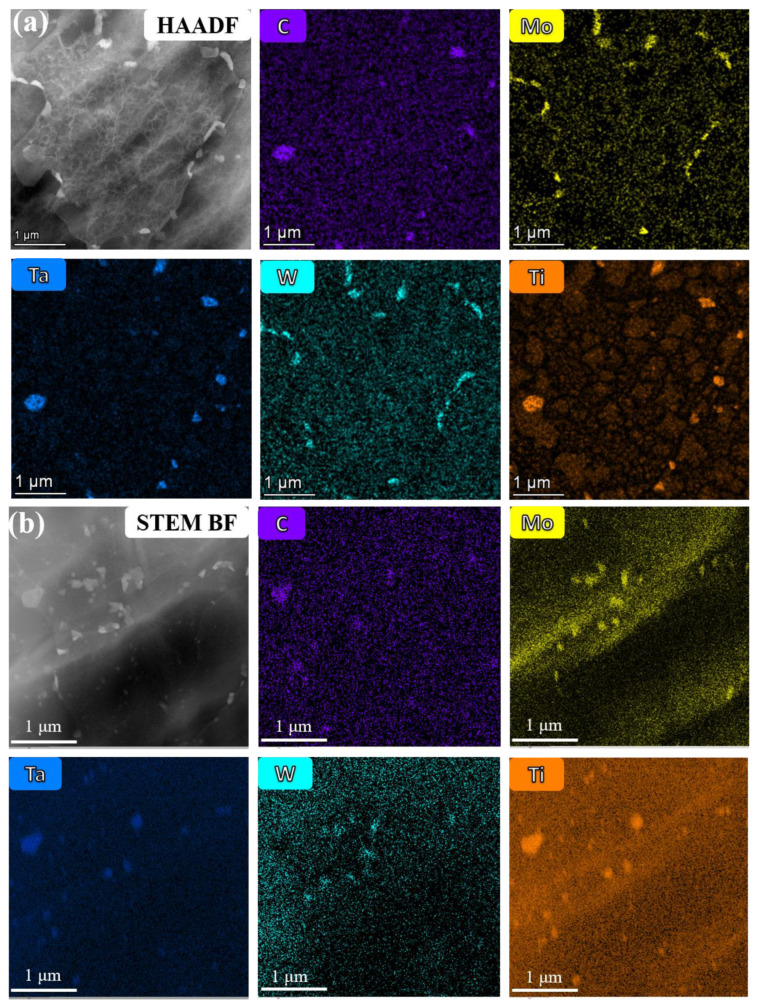
(**a**) HAADF-STEM image of the white precipitates and corresponding EDS mapping of 13Co. (**b**) Bright-field (BF) STEM image of the white precipitates and corresponding EDS mapping of 35Co.

**Figure 8 materials-16-02926-f008:**
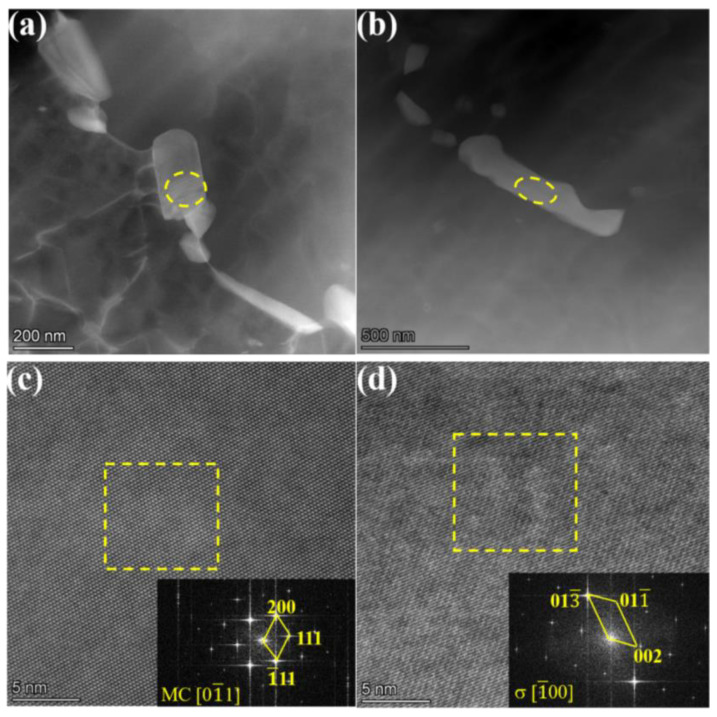
TEM investigation of the precipitates formed in the 13Co alloy. (**a**) HAADF-STEM image of MC carbides; (**b**) HAADF-STEM image of the σ phase; (**c**) HAADF-HRSTEM image and the corresponding FFT pattern of the MC carbide; (**d**) HAADF-HRSTEM image and the corresponding FFT pattern of the σ phase.

**Figure 9 materials-16-02926-f009:**
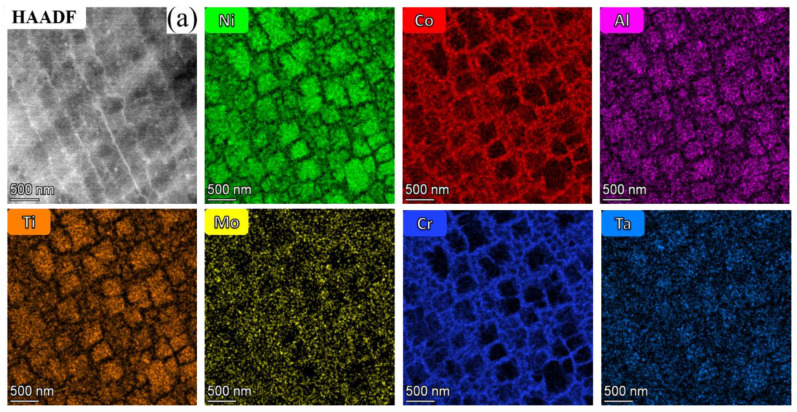
(**a**) HAADF-STEM images of the γ′ precipitates and corresponding EDS mappings in 13Co. (**b**) HAADF-STEM images of the γ′ precipitates and corresponding EDS mappings in 35Co.

**Figure 10 materials-16-02926-f010:**
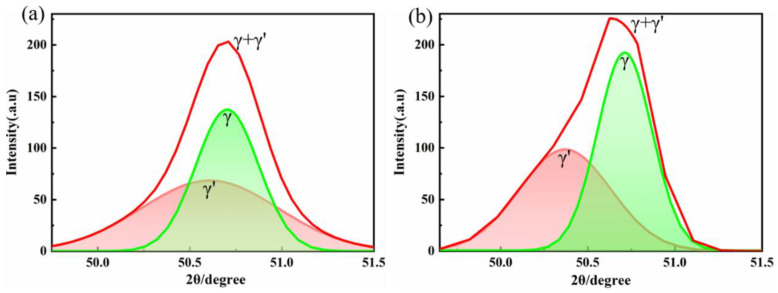
The (200) peaks of the XRD patterns for (**a**) 13Co and (**b**) 35Co with peak-fitted results with a Gaussian function.

**Figure 11 materials-16-02926-f011:**
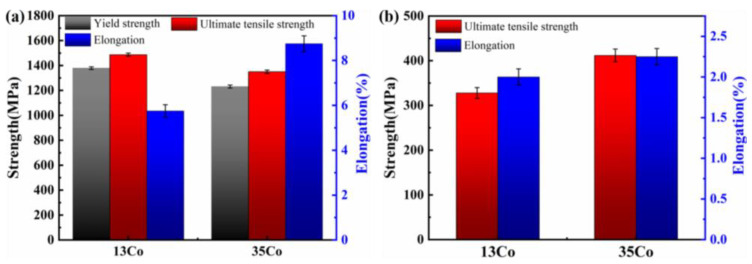
Mechanical properties of the 13Co and 35Co specimens after heat treatment at (**a**) room temperature and (**b**) 850 °C.

**Figure 12 materials-16-02926-f012:**
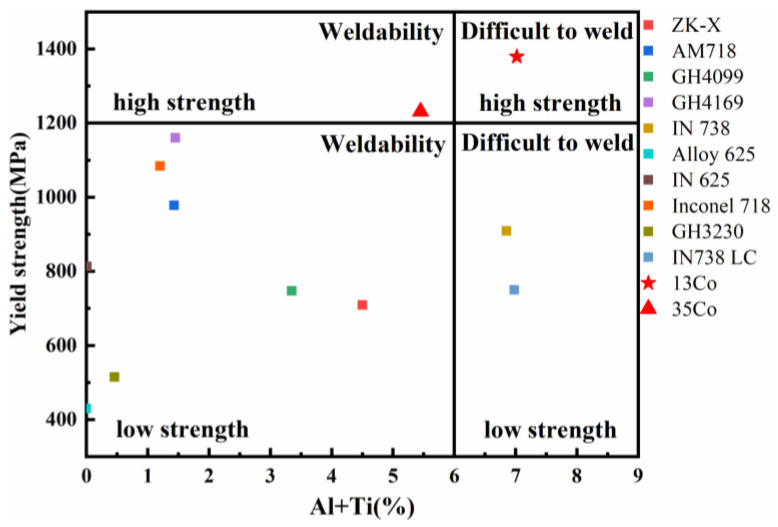
Comparison of the yield strength versus (Al + Ti)% for Ni-based superalloys in our work and other studies; all Ni-based superalloys were prepared via selective laser melting except for two superalloys, ZK-X and GH3230, which were prepared via laser melting deposition. Except for 13Co and 35Co alloys, the rest of the alloys are from the Refs. [39,40,41,42,43,44,45,46,47,48].

**Figure 13 materials-16-02926-f013:**
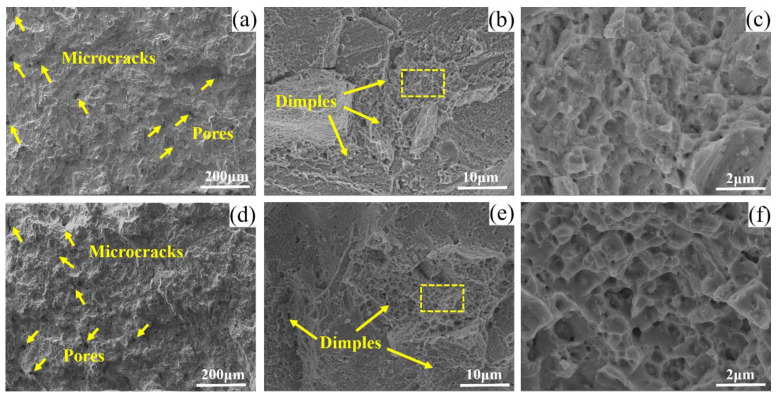
Room-temperature fractures of (**a**–**c**) 13Co and (**d**–**f**) 35Co.

**Figure 14 materials-16-02926-f014:**
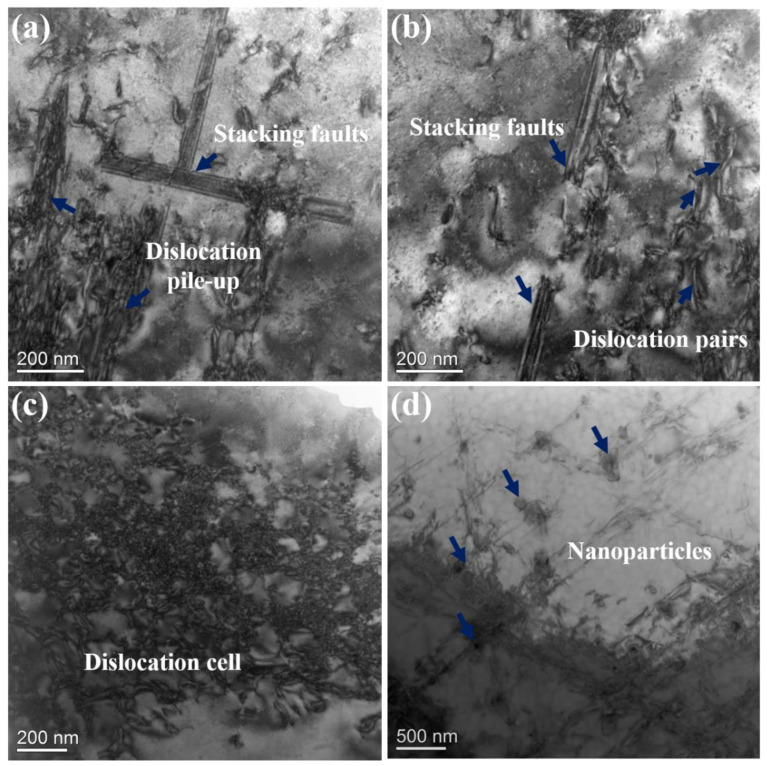
Deformed microstructure in the fractured specimen of 13Co: (**a**) stacking faults and dislocation pile-up; (**b**) stacking faults that sheared precipitates and interacted with dislocations; (**c**) dislocation cells; (**d**) nanoparticle-pinned dislocations.

**Figure 15 materials-16-02926-f015:**
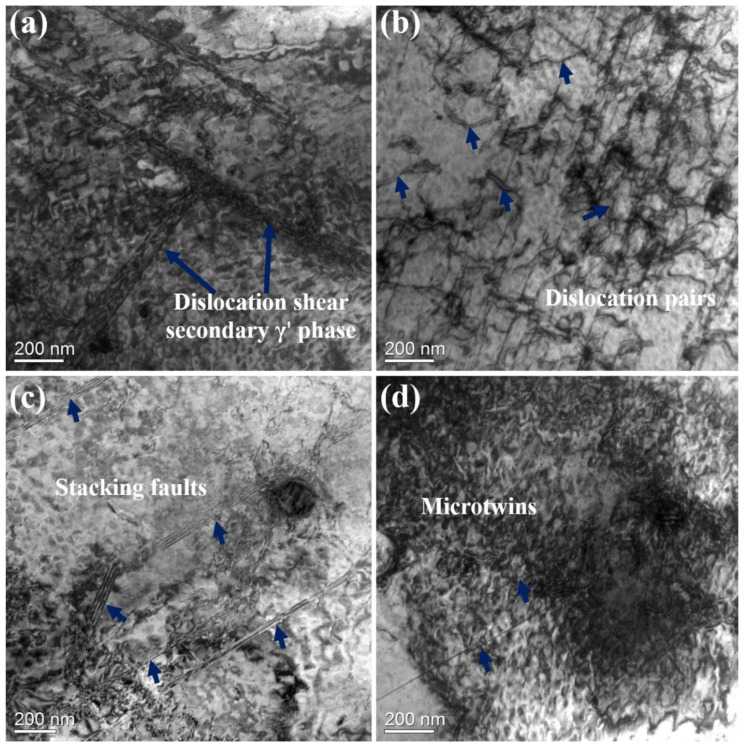
Deformed microstructure in the fractured specimen of 35Co: (**a**) dislocations that sheared secondary γ′ precipitates; (**b**) dislocation pairs within γ′ precipitates; (**c**) high-density stacking faults; (**d**) microtwins that cut through γ′ precipitates.

**Table 1 materials-16-02926-t001:** Nominal compositions of the two alloys (wt.%).

Alloy	Co	Cr	W	Mo	Ta	Ti	Al	C	B	Zr	Hf	Ni
13Co	13.00	11.96	4.03	4.02	4.02	4.03	2.99	0.06	0.04	0.03	0.14	Bal.
35Co	35.00	12.30	4.59	3.67	4.00	3.02	2.43	0.06	0.04	0.04	0.14	Bal.

## Data Availability

Not applicable.

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
