# Peer review of "The Effects of Co on the Microstructure and Mechanical Properties of Ni-Based Superalloys Prepared via Selective Laser Melting"

_materials, 2023, doi:10.3390/ma16072926_

Round 1

Reviewer 1 Report

The manuscript "The effects of Co on the microstructure and mechanical properties of Ni-base superalloys prepared by selective laser melting" shows some interesting results for introducing Co in Ni based superalloys. There are some points needs to be addressed.

1. Why 13% and 35% Co and why not a more analytical row with under 13% and over 35% Co. Are those numbers randomly selected or was a sense behind that? Hence 35 Co gave good results what about 50 Co?

2. In the introduction there should be some application given where such superalloys with higher strength are favorable. Please add those

3. The reviewer suggest to make supplementary as the results are many makes the manuscript a bit overloaded. Also some figures such as Figure 2a-d those need higher magnification as the mention cracks are not visible. Also Figure 6a the white precipitates are not visible. please provide better quality of this figures

4. 13 Co has better strength than 35 Co. The cracks are more pronounced in 13 Co, so why is 35 Co less strong? Please clarify this point

5. There are some abbreviations throughout the manuscript which are not defined. Page 1 line 35 and 36: Hf, CM247LC, IN738LC. what means such? Page 2, line 65: D50 please define. Figure 12, the legends have different names. What are those such as ZK-X and GH3230? please give some explanation in capture of the Figure.

Reviewer 2 Report

The paper “materials-2285661” related to laser cladding was reviewed. Please follow the comments carefully and resubmit your paper for the next consideration and reviewing process.

1.     How did you select the meltpool in Figure 2 C?

2.     The research methodology is not clear. Please add more information about the methodology.

3.     The size of the font in Figure 3 “IPF maps” is too small.

4.     If possible, compare your results with the same literature. If no literature is available, please add a statement and mention that this is the uniqueness of the current paper.

5.     What are the governing factors provided in the discussion? This is important to bold this section for a better understanding of the readers.

6.     The paper has some typos. The authors need to proofread the paper.

7.     The new and emerging technology which is called “additive manufacturing” has some pros compared to machining and other conventional manufacturing methods. This needs to be highlighted in your paper. Refer to the following paper and add it to the introduction. “Laser subtractive and laser powder bed fusion of metals: review of process and production features”

8.     The absorptivity of laser in additive manufacturing drives many factors such as meltpool morphology. Refer to the following paper to highlight this note in your manuscript. “The effect of absorption ratio on meltpool features in laser-based powder bed fusion of IN718”.

Reviewer 3 Report

materials-2285661

1. The authors need to explain the novelty of the current work.

2. Self citations and citations from one country should be avoided.

3.The plagiarism levels are very high as can be found in the attached report. This need to be bought down to less than 20%.

4. Why only 13 wt.% and 35 wt.% Co  were chosen? Why not other compositions?

5. Figure 1. SEM images of the powder morphology for (a) 13Co, (b) 35Co. Both should be same in powder form. Why are authors providing two separate SEM images? Is it after compaction? Please clarify.

6. Figure 1.(d) the geometry of the tensile specimen with a thickness of 2 mm. Please confirm if all other dimensions are in mm and indicate the same. What is the ASTM standard for this dimensions?

7. Please explain the difference of the current work and the previous work cited below:

[20] O. Adegoke, S.R. Polisetti, J. Xu, J. Andersson, H. Brodin, R. Pederson, P. Harlin, Influence of laser 491 powder bed fusion process parameters on the microstructure of solution heat-treated nickel-based superalloy Alloy 247LC, Materials Characterization, 183 (2022) 111612-111628.

[21] R. Engeli, Selective laser melting & heat treatment of γ´ strengthened Ni-base superalloys for high  temperature applications, ETH Zurich Research Collection, (2017).

8. Please add an error bar (standard deviation) in Figure 11. (both figs)

9. What do the authors infer from Figure 10. Is it really required? What conclusions are made from Fig. 10. Please explain.

10. What do the authors mean by ‘white precipitate’ in Fig. 6?

11. Reference format can be checked.

12. The authors can cite the following paper and improve their discussion on laser interaction: https://doi.org/10.1007/s13369-022-07256-9, https://doi.org/10.1016/j.optlastec.2022.108210

Round 2

Reviewer 3 Report

The authors have addressed the comments meticulously. I now recommend acceptance.